# Deep Learning Unravels Differences Between Kinematic and Kinetic Gait Cycle Time Series from Two Control Samples of Healthy Children Assessed in Two Different Gait Laboratories

**DOI:** 10.3390/s25010110

**Published:** 2024-12-27

**Authors:** Alfonso de Gorostegui, Damien Kiernan, Juan-Andrés Martín-Gonzalo, Javier López-López, Irene Pulido-Valdeolivas, Estrella Rausell, Massimiliano Zanin, David Gómez-Andrés

**Affiliations:** 1PhD Program in Neuroscience, Universidad Autonoma de Madrid-Cajal Institute, 28029 Madrid, Spain; alfonso.degorostegui@gmail.com; 2Department of Anatomy, Histology & Neuroscience, School of Medicine, Universidad Autónoma de Madrid (UAM), 28029 Madrid, Spainestrella.rausell@uam.es (E.R.); 3Movement Analysis Laboratory, Central Remedial Clinic, Clontarf, D03 R973 Dublin, Ireland; dkiernan@crc.ie; 4Escuela Universitaria de Fisioterapia de la ONCE, Universidad Autónoma de Madrid, 28034 Madrid, Spain; jumago@once.es; 5Department of Rehabilitation, Hospital Universitario Infanta Sofía, Fundación para la Investigación e Innovación Biomédica del Hospital Universitario Infanta Sofía y Hospital del Henares, San Sebastián de los Reyes, 28702 Madrid, Spain; jlopezlo@salud.madrid.org; 6Departamento de Medicina, Salud y Deporte, Universidad Europea de Madrid, Alcobendas, 28102 Madrid, Spain; 7Instituto de Física Interdisciplinar y Sistemas Complejos IFISC (CSIC-UIB), Campus UIB, 07122 Palma de Mallorca, Spain; 8Pediatric Neurology, ERN-RND, Euro-NMD, Vall d’Hebron Institut de Recerca (VHIR), Hospital Universitari Vall d’Hebron, Vall d’Hebron Barcelona Hospital Campus, Passeig Vall d’Hebron 119-129, 08035 Barcelona, Spain

**Keywords:** gait, deep learning, external validity, children

## Abstract

We investigate the application of deep learning in comparing gait cycle time series from two groups of healthy children, each assessed in different gait laboratories. Both laboratories used similar gait analysis protocols with minimal differences in data collection. Utilizing a ResNet-based deep learning model, we successfully identified the source laboratory of each dataset, achieving a high classification accuracy across multiple gait parameters. To address the inter-laboratory differences, we explored various pre-processing methods and time series properties that may have been detected by the algorithm. We found that the standardization of the time series values was a successful approach to decrease the ability of the model to distinguish between the two centers. Our findings also reveal that differences in the power spectra and autocorrelation structures of the datasets play a significant role in the model performance. Our study emphasizes the importance of standardized protocols and robust data pre-processing to enhance the transferability of machine learning models across clinical settings, particularly for deep learning approaches.

## 1. Introduction

Gait analysis plays a growing role in clinical decision-making in pediatric motor disorders, including diagnostic orientation, disease progression monitoring, and therapeutic decision support [1]. Children with conditions such as cerebral palsy [2], muscular dystrophies [3], spinal muscular atrophy [4] or pediatric motor disorders [5,6] often exhibit abnormal gait patterns that can be assessed with instrumental gait analysis and in which biomarkers can be extracted [7]. However, despite its clinical value, the widespread application of gait analysis in pediatric clinics is often hindered by the complexity of interpreting high-throughput, multidimensional data. Gait signals—such as joint angles, ground reaction forces, and muscle activations—are typically highly correlated and require expert knowledge for accurate interpretation, making it difficult to fully harness the potential of gait analysis in routine clinical care and clinical research [8].

Machine learning (ML) offers a promising approach to address this challenge. By automating the interpretation of complex data, machine learning models can help reduce the burden on clinicians and make gait analysis easier and faster to interpret [9,10]. In particular, deep learning (DL) has shown potential in processing time series data such as gait signals, extracting meaningful patterns without needing a priori feature selection [11]. Moreover, DL can easily incorporate complex data from other sources and has interesting properties such as transfer learning, which may be critical in generation models for rare disorders in pediatric neurology [12].

However, a key challenge remains in translating ML models from one clinical setting to another. Gait data collected in different laboratories may differ due to variations in sensor setups, measurement protocols, or participant demographics, complicating the generalization of trained models. Understanding how deep learning models for time series data perform when comparing similar datasets collected in different laboratories is critical, especially in pediatric populations, where precision is essential. This study explores this challenge by comparing gait data from two groups of healthy children, measured in different laboratories, using state-of-the-art deep learning techniques. The objective is two-fold. On the one hand, we want to understand whether the two datasets are equivalent, not only under a qualitative visual inspection, but also from a quantitative DL perspective, or, in other words, if DL models could use these datasets interchangeably. On the other hand, we analyze which characteristics of the data are different, in spite of using shared equipment and assessment protocols, to provide guidelines to improve future experimental settings.

## 2. Methods

### 2.1. Patients

We included gait data from healthy subjects assessed in two different laboratories. The first group of patients was evaluated in the Central Remedial Clinic in Dublin and included 96 children aged from 4 to 16 years who had been determined as healthy children after careful medical history retrieval and examination. Collection of retrospective data of healthy subjects previously evaluated in the laboratory was allowed by the local ethics board (CRC Ethics approval 12116) and informed consent for participation was not required as per local legislation (HSE National Policy for Consent in Health and Social Care Research (V1.1, 2023)). The second group of patients was evaluated in the clinical gait analysis laboratory in “Escuela de Fisioterapia de la ONCE-UAM” and was formed of 31 children aged from 5 to 15 years who had been considered as healthy children after a systematic medical history and examination. Inclusion criteria were as follows: adequate schooling, absence of any clinical history of neurological, cardiovascular, or systemic diseases, absence of uncorrected visual or hearing impairment, absence of known orthopedic pathologies in the previous six months, and negative results after a screening of unknown orthopedic pathologies through the exploration protocol “Scottish Rite Hospital” [13] and a neurological examination. Research was approved by the local ethics commitee (Universidad Autónoma de Madrid, CEI-27-685).

### 2.2. Gait Methods and Laboratories

A Codamotion system (Charnwood Dynamics Ltd., Rothley, UK) was used for gait evaluation. In accordance with a manufacturer-designed anthropological segment model, light-emitting markers were affixed to reference locations of the subjects’ legs. Signals were captured at 200 Hz during the task. Data from a Kitsler power platform, which the manufacturer also incorporates into the gait system, were used to determine the ground response force and inverse kinetics. Following their first training, the children were encouraged to travel a long walkway path multiple times at their own impulsive, natural pace. It was requested that the walkway be repeated eight to twelve times by the children. Continuous real-time kinematic data were collected by the system during every full walkover. Turns were excluded from the analysis. After the acquisition session, individual gait cycles were isolated, by manually marking their beginning (heel contact) and their end (next heel contact of the same foot). Next, each selected cycle was again reviewed to check the consistency of the signal reception. Post-processing was performed with custom-made protocols in the software provided by the gait system manufacturer and was strictly limited to data extraction and preparation. For every side gait cycle, we studied the kinematic time series (100 time epochs) of the 3 angular planes (sagittal, horizontal, and coronal planes) from 5 joints (pelvis, hip, knee, ankle, and forefoot). We also studied the mediolateral, anteroposterior, and sagittal ground reaction forces (GRFs) and the estimated moment and power time series for the sagittal plane in hip, knee, and ankle joints. For each subject, up to five left and up to five right gait cycles were selected according to technical quality criteria.

Central Remedial Clinic laboratory (Dublin) uses a 14 m long platform, which is placed in a larger room (Figure 1). The gait laboratory in “Escuela Universitaria de Fisioterapia de la ONCE Universidad Autónoma de Madrid (Figure 2) has an 8 m long walking platform integrated into a larger room. The space after the platform is smaller, likely suggesting that the average walking speed is slower, step cadence is decreased, and the stance time is prolonged in Madrid’s healthy children compared to their Dubliner peers (see Table 1).

### 2.3. Deep Learning Classification Tasks

Many alternative architectures have been presented in the literature in the last decade, proposing different ways of constructing deep learning models for the classification of time series. In most cases, these have been adapted from similar problems, e.g., image classification, but are still able to yield excellent results [14], even above those achievable with more traditional approaches [15].

In this work, we specifically leverage Residual Networks (ResNets), a type of convolutional architecture introduced to improve the training of deep networks [16]. The starting points are Convolutional Neural Networks (CNNs), i.e., artificial neural networks exploiting hierarchical patterns that may be present in the data and based on convolutional operations; a filter (also called a kernel) slides along the input time series, performing a matrix multiplication element after element [17]. Being designed for processing sequential data, CNNs are able to efficiently capture local patterns and features. The ResNet model is a variation of CNNs that addresses the vanishing gradient problem often encountered in very deep networks. Specifically, it introduces shortcuts, that is, links between two distant layers without involving the intervening ones. The result is a block structure, in which each block is a set of layers in which connections are sequentially present between every set of shortcuts.

ResNet models have been chosen in this work due to their performance, usually higher than other DL architectures, albeit at the expense of a higher computational cost [15]. Key hyperparameters in this configuration include the number of residual blocks, set to 5 in this case, with each block consisting of 5 layers that contain 32 filters of size 10. Training has been performed over 200 epochs—i.e., the number of times the full training dataset is presented to the model. An in-house software implementation has been built in Python, version 3.11, using standard libraries including TensorFlow [18] and Keras [19].

In order to ensure the generalisability of the results presented herein, and exclude their dependence on a specific architecture, some key ones will also be replicated using additional DL models. These have been chosen among those being considered the best ones in the literature; for instance, the interested reader may refer to Ref. [14] for an in-depth description and comparison. Model configurations and hyperparameters have been taken from those optimizing similar classification problems, e.g., as analyzed in [15]. For the sake of completeness, basic descriptions are included in what follows, while more details can be found in the corresponding referenced works.

*Multi-Layer Perceptrons* (MLPs) [20]. Basic feedforward neural networks composed of two hidden layers of 320 nodes each, all of them fully connected to the units of the next layer.*Convolutional Neural Networks* (CNNs) [17]. Basic convolutional architectures, i.e., without layer skipping used in ResNet. We used a configuration with two hidden convolutional layers with 24 filters per layer, with a kernel size of 10 for each filter.*Transformers* [21]. These models are mostly used for processing textual data. They are based on encoding groups of consecutive values into vectors, which are then analyzed by “attention heads”, or small structures that evaluate the importance of one element with respect to neighboring ones. Key hyperparameters include the number of Transformer blocks, the attention head count, and the embedding dimension—all of them set to 4.*Long Short-Term Memory* (LSTM) [22]. This is a type of recurrent neural network that is capable of learning long-term dependencies in sequential data, by using memory cells and gating mechanisms to control information flow. Here, we included two hidden layers of 128 units each, and a drop-out rate of 0.5.

Each classification problem considered here starts with a set of univariate time series representing an individual kinematic or kinetic parameter, and, thus, a specific aspect of the gait along one cycle. No pre-processing beyond what will be described in Section 3 was initially performed to keep the data as close as possible to those originally recorded; specifically, the time series have not been amplitude-normalized, as the values were mostly within the range (−2, 2). An independent ResNet model has been trained on data from the two datasets, with the objective of identifying the source laboratory. In order to simplify the interpretation of results, all classification tasks use the same number of time series for both groups; time series in the over-represented group are selected at random. The performance of the model has then been estimated over a new set of random time series, using the accuracy metric (i.e., fraction of correctly classified time series, here also denoted as “classification score”). To account for the natural stochasticity of the training process, this has been repeated 100 times, with the final classification score corresponding to the median of the individual accuracies.

The data presented in this study are available on request from the corresponding authors. The data are not publicly available due to privacy and ethical restrictions.

## 3. Results

### 3.1. Time Series Discrimination

We start the analysis by training DL models using the raw time series recorded by both laboratories, aimed at discriminating (and hence, predicting) the origin of each time series. Figure 3 reports the obtained classification scores, organized by types of signals (see the three colors). As previously discussed, each joint/angular plane has been analyzed independently from the others in a univariate classification task. Except for the kinematics of the pelvis and hip, the ResNet models can identify the origin laboratory with very high accuracy, in some cases reaching 100% success. In the knee, ankle, and foot kinematics, although the classification scores were always large, better results were found in the transverse and rotational planes than in the sagittal one. While part of this success is due to the sensitivity of the ResNet model, similar results can be obtained using other (less powerful) DL models; see Figure A1.

We utilized two cross-validation methods—leave-one-cycle-out (LOCO) and leave-one-patient-out (LOPO)—to evaluate the ResNet deep learning model’s generality to classify gait data across different time series features. LOPO cross-validation selectively assesses the impact on model generalization of inter-subject variability while LOCO does not consider the origin of the variability. Only mild differences in classification scores were found using the LOCO and LOPO approaches (see Figure A2). We additionally evaluated whether the differences between the two groups could be due to the resolution of the time series (see Figure A3). While lowering such time resolution can help reduce the classification score, this does not lead to major improvements unless a large part of the information is deleted.

To understand what characteristics the DL can use to discriminate between the two laboratories, we present in Figure 4 an example of three time series extracted at random for the moment of the hip, corresponding to Dublin’s (top panel) and Madrid’s (bottom panel) datasets. Two key differences stand out: the initial part of the time series is not similar, with Madrid’s time series displaying larger fluctuations; Madrid’s time series also seem to be more variable and unpredictable, with the presence of high-frequency content.

These major differences can easily be eliminated. On the one hand, the first 30 points of all the time series can be deleted (eliminating the first double support and initial part of the single support gait phase), thus only considering the final 70% of them. On the other hand, the time series have been smoothed using a Savitzky–Golay filter [23], using a third-order polynomial and a window size of 9 points. Finally, to discard trivial amplitude differences, every time series has been rescaled to have zero mean and unity standard deviation. The time series of both laboratories become much more similar after this processing; see Figure A4 for a graphical representation of the outcome of processing the time series in Figure 4. Despite this, the ResNet models trained thereon still achieve a very high classification score, with only a minimal decrease across all the signals—see the black bars of Figure 5 for the reduction in the score, and Figure A5 for the full results.

### 3.2. Differences in Value Distributions

As is common when dealing with real-world data, the distribution of the values in the different time series for both groups of healthy children is not exactly the same. For instance, in Figure A6, we show the particular case of the hip moment time series. To assess whether ResNet could be leveraging these differences in the distribution of time series values, we performed a standardization aimed at deleting the influence of these inherent disparities and obtain a more unified data structure. Specifically, the values of each time series obtained in the previous step (hence, being a length of 70) have been renormalized between zero and one, such that the *n*-th smallest value is mapped to the value n/70. Note that the result is sets of time series with the exact same value distribution.

This strategy leads to a significant decrease in the accuracy of the models, particularly in those relative to kinematic information (see golden bars of Figure 5 for the reduction in the classification scores). Nevertheless, classification scores remain substantially high (higher than 0.7 or 0.8) in many time series (Figure A7 for classification scores before and after standardization of time series values).

### 3.3. Differences in the Autocorrelation Structure

One of the key elements detected by DL models are local structures of autocorrelation in the time series; to illustrate, the key component of a ResNet model is a convolutional operation, which calculates the presence of short-scale correlations in the data. To check for differences between the time series of both laboratories, Figure A8 reports the median evolution of the autocorrelation of each signal, calculated on the time series modified in the previous step. The global structure is similar in all cases: starting from positive autocorrelations for small values of the lag τ, the value has a local negative minimum at around τ=30, finally converging towards a zero autocorrelation. At the same time, some major differences can be seen near the minimum, both in amplitude and location.

To understand whether these differences are at the basis of the classification performed by the DL models, we plot in Figure 6 three scatter plots, respectively, of three metrics as a function of the classification score:Δ minima (left panel), i.e., the difference between the τ corresponding to the minima in both sets of time series. This metric is calculated as $|\log_{2}\frac{\tau_{Madrid}}{\tau_{Dublin}}|$, where $\tau_{Madrid}$ and $\tau_{Dublin}$ are the location of the two minima.Δ area (central panel), calculated as the area between both autocorrelation curves.Continuous Ordinal Patterns (COPs). These are a generalization of the traditional ordinal pattern approach [24], in which not only the relative order of the data points but also their magnitudes are taken into account, and have been proven to be relevant in characterizing time series, assessing irreversibility [25], and improving causality tests [26]. In more detail, random COPs were applied to the time series, and the difference between the result obtained in each group was measured through Cohen’s *D*. The higher this difference, the more the structures of the correlations within the time series are different; hence, this metric can be seen as a generalized version of the previous ones.

As can be seen in Figure 6, the higher the difference between the autocorrelation structures, the higher the classification score yielded by the ResNet model—see also the best linear fit and the corresponding *p*-value in each panel, as well as the Spearman rank correlation.

Note that the presence of negative values in the autocorrelation function is a consequence of the oscillatory nature of the movement, something that can easily be spotted in the examples of Figure 4. The changes observed in Figure A8 and validated in Figure 6 thus indicate differences in the amplitude and frequency of such oscillations.

### 3.4. Differences in Frequency Content

The previous results on the autocorrelation structure of time series also suggest the presence of differences in their frequency content. To test this, the power spectrum of each time series has been estimated using Welch’s method [27]. Compared to other standard approaches (e.g., the Fast Fourier Transform), Welch’s approach yields more stable results, especially in the presence of noise and non-stationarities.

As in the case of autocorrelation, a full depiction of the median power content of all the time series is included in Figure A9; on the other hand, Figure 7 provides an overview of the relationships between the obtained classification score, and the Δ maxima (left panel, i.e., the difference between the maxima in the two power spectra) and the Δ area (right panel, difference between the power spectra). A clear relation can be appreciated in the latter case—see linear fit and *p*-value in the corresponding panel, as well as the Spearman rank correlation. The spectra’s maxima, on the other hand, seem to yield more limited information; nevertheless, this may be caused by the extreme nature of the maxima as parameters and the limited length of the time series, which reduces the available resolution of the spectra themselves. The relationship between the classification scores and the area of the power spectra reveals that the model may be focusing on specific variations and oscillations that are occurring at different frequencies in the two datasets.

### 3.5. Localization of Differences

To assess the contribution of different gait cycle segments to the classification performance, we divided the time series into sub-windows including information from 20% of the gait cycle. We excluded the first 30% of the gait cycle due to the presence of evidence differences in this portion of the gait cycle. In Figure A10, we show that most of the sub-windows retain significantly high classification scores, comparable to the results of using the whole gait cycle. Figure 8 shows a summary of these results and highlights the sub-windows with higher classification scores. Notice that there is not a predominant section of the gait cycle which may be contributing more to the laboratory classification.

## 4. Discussion and Conclusions

In this study, we demonstrate that deep learning can accurately classify gait data from two samples of healthy children that were evaluated in two different laboratory settings that share equipment and assessment approaches. Even though laboratories share many conditions, ResNet was able to detect the origin of the data with significant accuracy in every time series (see Figure 3), probably based on the combination of small differences. The implications of this result can hardly be overstated, as the equivalence of different datasets is an essential prerequisite for generalizability. In other words, if two datasets are highly differentiable by a DL model, it also implies that a DL model trained on one of them will not be reliable when applied to the other one. We used different validation procedures and other machine learning approaches with robust (but significantly less accurate) results (see Figure A1). Some differences are easy to spot by just looking at the time series. These include the difference in the initial part of the time series, and the presence of a higher high-frequency noise in one of the datasets (see Figure 4). We eliminate these two aspects yielding minor differences in the classification score; see Figure A5. Lowering the temporal resolution of the time series leads to a reduction in the classification ability, but in contest to our expectation, the models’ accuracies remain particularly robust (see Figure A3). This robustness to time series modifications and the superiority against other machine learning algorithms highlights the power of deep learning in detecting laboratory-variant features. As a valid pre-processing approach, we studied the standardization of the distribution of isles in the time series. When time series are modified to have a uniform distribution, classification scores substantially drop, especially for moment and kinematic variables (see Figure A7). Finally, we have shown that the classification score is strongly related to differences in the autocorrelation structures and the power spectra of the signals. Specifically, the former ones seem to indicate that the oscillatory nature of the signal is different in the two laboratories (see Figure A8), and contrary to other differences, this is difficult to reduce through processing after data collection. We also explored the part of the gait cycle that is more relevant for distinguishing between the two laboratories without any clear pattern.

We have recorded two sets of gait data in two laboratories, trying to match all conditions as closely as possible. In contrast to our initial expectations, in which a detectable but small difference would be found, the model was able to accurately distinguish between the groups. The two laboratories use the same gait system but there are some differences in the space and the floor the children walked upon. There were also small differences in the temperature and illumination. These variations provoked differences in the walking strategy, as shown not only by the distinct spatiotemporal parameters but also the recording being affected, introducing distinct features in the gait signals. We believe that these differences in the spatiotemporal configuration of gait leads to differences in the analyzed time series. As the spatiotemporal parameters are different but overlapping between the two healthy datasets, this could partially explain the ability of the models to detect differences. The walking speed is the spatiotemporal parameter whose influence on kinematic and kinetic time series has been studied more [28]. For instance, slower speeds typically increase support times and alter muscle activation patterns, as the body adjusts to maintain stability under different constraints. At lower speeds, stance phases dominate, extending the duration of double support phases. This adaptation could result in less dynamic, stability-focused muscle activation, leading to more gradual shifts in joint movements and ground reaction forces. Conversely, as the speed increases, the stance and double support times shorten, demanding faster and more intense muscle activations to propel the body forward, changing the timing and intensity of joint power generation [29].

The presence of higher noise at the the beginning of the time series could arise from differences in the gait walkway used by the children, such as the elasticity of the platform, distance, and different self-selected walking speeds. Another source of bias may be related to the different strategies for selecting valid gait cycles, a process that was carried out independently in the two centers. We think that the impact of this fact is minimal after sharing the criteria we used when two of the Spanish researchers enjoyed a short research stay in Dublin.Moreover, small differences in signal filtering in the different sensor settings may also contribute to increased inter-laboratory differences. Another source of difference between the groups could have been provoked by the different cultural backgrounds [30] and stimuli [31] that the healthy children may have received during the data acquisition.

These differences between controls measured at different laboratories impact the data transferability. We believe that the strict standardization of protocols remains critical to eliminate the influence of equipment differences and the walking environment. However, there are some limitations to implementing this in real life. One potential solution would be the use of virtual reality or treadmills, but this also modifies the gait performance, decreasing the natural validity of the studies [32,33]. Another possible measure for easier inter-laboratory transferability of the data and models could be the improvement of calibration moving from particular sensors (easy and currently performed) into a calibration of the whole system (complex and highly unexplored) [34]. These limits point out the importance of increasing cross-validation studies and the application of data harmonization techniques, which will be more powerful if fed with more lab-to-lab comparisons.

Our results also highlight how complicated the translation of models trained in one laboratory or environment into a different one can be. This limits the deployment of machine learning models in real-world healthcare environments, which is critical to provide a more personalized and accurate gait analysis to children and other patients [35,36]. We think that accumulating information about the differences between different control datasets coming from different laboratory settings and about how models may detect differences between control groups is one of the first steps to ease the applications of models trained in one or several labs into different ones.

Approaches in pre-processing data (simple as standardization or more complex as data harmonization with autoencoders [37,38]) or evaluating the impact of using laboratory-invariant features in the early part of model development could be developed to overcome this problem [39] . This step allows models to focus more on meaningful gait patterns rather than dataset-specific distinctions, thereby improving generalization and potentially leading to a more robust model performance. In our study, we analyzed different approaches. The most successful one was the standardization of the distribution of the time series values. Another promising strategy to address this challenge is transfer learning, a technique where a model trained on one task is adapted to perform on a related task or environment, such as a different laboratory [40].

The observation that differences in the power spectra relate to the performance of the deep learning model suggests that spectral features play a significant role in the model’s ability to distinguish between datasets. The power spectrum of a signal captures how the signal changes are distributed across different frequencies [41]. So, different power spectra imply distinct frequencies in which normal kinematic and kinetic changes occur. This can occur due to several reasons, such as differences in the sampling rates (not likely in our case), and different sensor noises (maybe related to differences in the floor material and/or other conditions in the laboratory), or intrinsic biomechanical differences (for instance, those provoked by different speeds, cadences, and distributions of support and swing time in the gait cycle of the two datasets). When two datasets have different power spectra, this implies differences in the frequency components of gait patterns—possibly due to variations in sampling rates, sensor noise, or intrinsic biomechanical differences. These spectral differences can introduce distinct features or biases that the deep learning model learns to associate with specific datasets.

The combination of two cross-validation approaches is also a relevant contribution to our work. The LOPO cross-validation results reveal a more pronounced challenge in generalizing across different individuals. As expected, the classification scores tend to be lower than those observed in the LOCO setting. This outcome is consistent with the notion that gait patterns can vary significantly between individuals due to factors such as body morphology, walking habits, and subtle differences in movement strategies. The comparison between LOCO and LOPO highlights the strengths and challenges of applying deep learning models to gait analysis. In LOCO, the high classification scores suggest that the ResNet model performs well in capturing intra-subject gait variability, which appears to be the most stable and informative. However, in LOPO, where the model is tasked with generalizing across different individuals, we found lower classification scores but cross-validation classification scores remained high. This reveals that the ML approach can detect intra-individual variability, but mostly inter-individual variability.

Deep learning has been extensively used in gait analysis [11]. Deep learning approaches have shown that they are able to demonstrate even individual gait patterns and predict identities in healthy subjects [42] or accurately predict the kinematic performance under different circumstances [43]. This kind of application suggests that deep learning could make classifications with small differences in the time series.

Our study has limitations. We have only tested a single deep learning algorithm. In our experience, ResNet is one of the most powerful available approaches for time series, and the fact that a single model performs in the extremely fine way in which ResNet performs is already a cause for concern in the application of deep learning models without adequate data harmonization. Moreover, we tested other ML approaches which also show good accuracy for classification. We only included two laboratories, but in this way, we were able to compare two laboratories with very similar settings. Even in these circumstances, we were able to find differences. These results highlight the need for studies including more laboratories and different gait analysis systems. Another limitation, which is intrinsic to the use of deep learning approaches, is the problems we have in explaining which features are driving the detection of differences. However, we think that we have explored several explanations.

In conclusion, our study demonstrates that deep learning and other machine learning algorithms can accurately detect differences between two samples of healthy children assessed in two different laboratories. Some pre-processing techniques may reduce the classification ability of the algorithms, and may be important in the transfer of applications for supporting clinical decisions between laboratories.

## Figures and Tables

**Figure 1 sensors-25-00110-f001:**
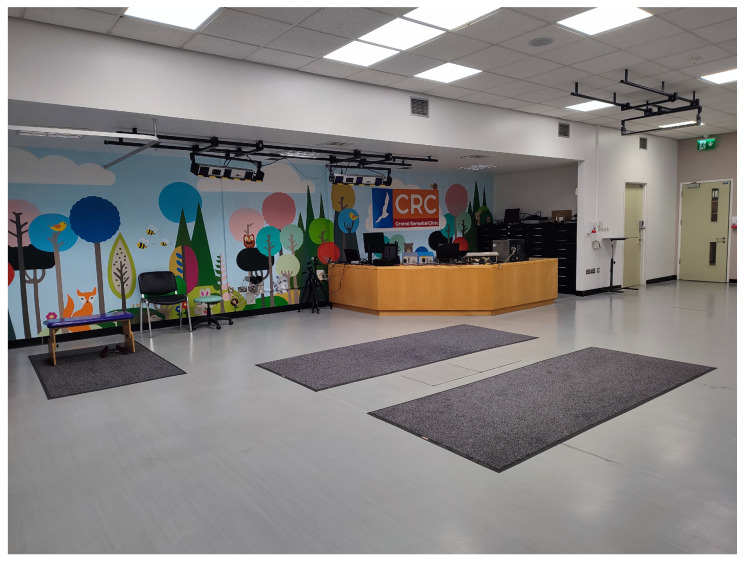
Image of the CRC gait laboratory in Dublin.

**Figure 2 sensors-25-00110-f002:**
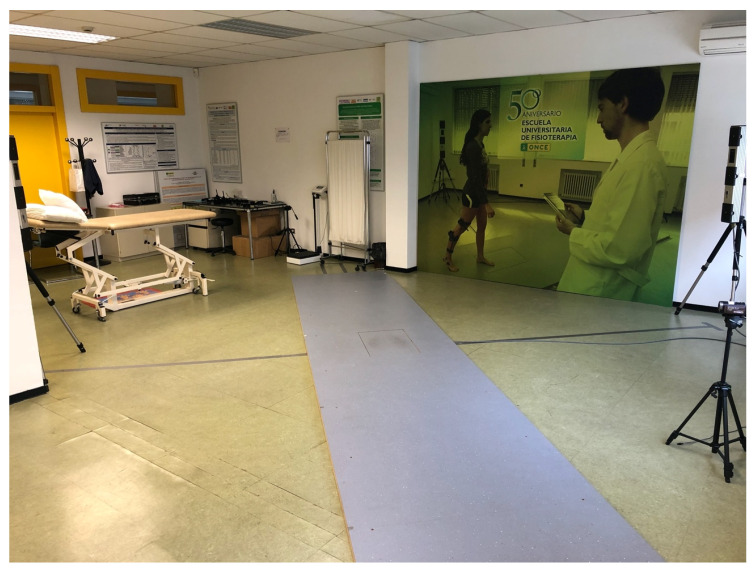
Image of gait laboratory in Escuela de Fisioterapia de la ONCE (Madrid).

**Figure 3 sensors-25-00110-f003:**
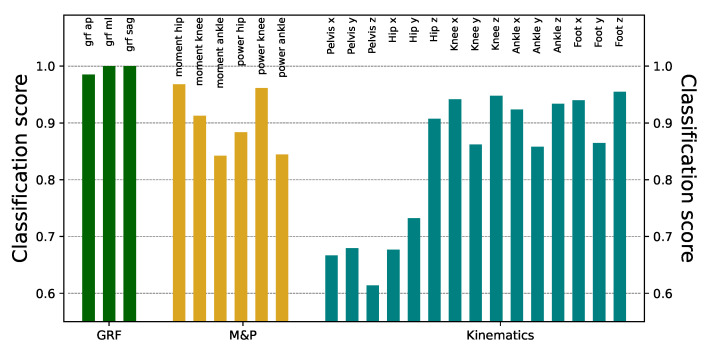
Classification scores obtained from the raw time series. Scores have been obtained using ResNet models, and correspond to the median over 100 independent realizations.

**Figure 4 sensors-25-00110-f004:**
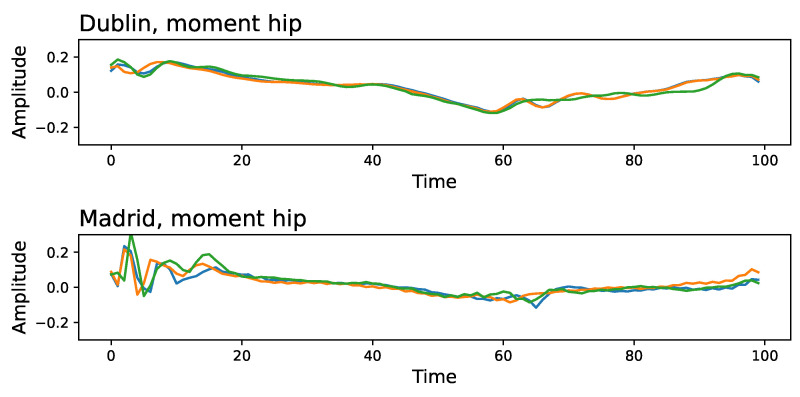
Three examples of time series (see different colours), corresponding to the moment of the hip of three different subjects, as recorded in Dublin (top panel) and Madrid (bottom panel).

**Figure 5 sensors-25-00110-f005:**
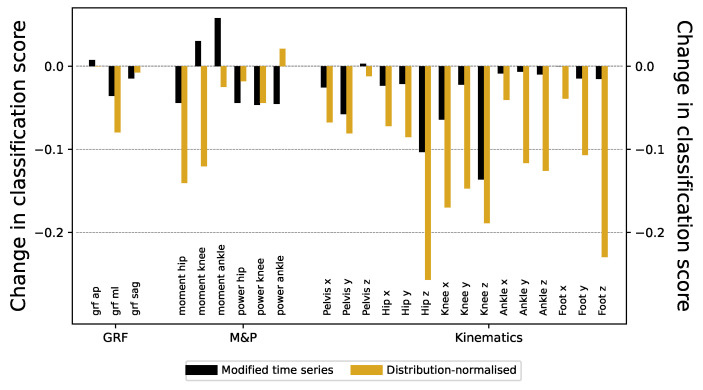
Changes in the classification score of time series under different modifications. Black and golden bars depict the drop observed using ResNet models for different time series, respectively, when considering the modified ones of Section 3.1 (black bars), and the ones with their distribution modified as per Section 3.2 (golden bars). For full results, see Figure A5 and Figure A7.

**Figure 6 sensors-25-00110-f006:**
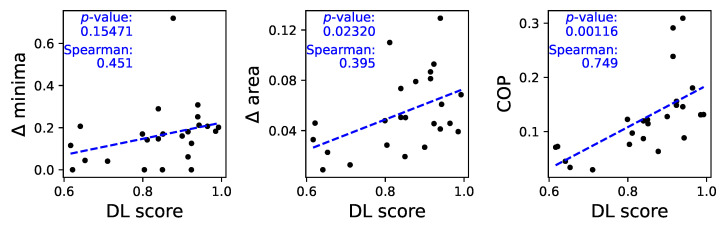
Analysis of the autocorrelation structure of time series. Left and center panels, respectively, report the Δ minima and the Δ area as a function of the classification score. The right panel reports the evolution of the COP as a function of the same score—see main text, Section 3.3 for definitions. See also Figure A8 for full results.

**Figure 7 sensors-25-00110-f007:**
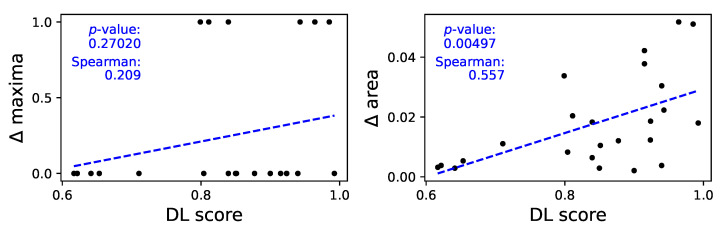
Analysis of the power spectrum of time series. Left and right panels, respectively, report the Δ maxima and the Δ area as a function of the classification score—see main text, Section 3.4 for definitions. See Figure A9 for full spectra.

**Figure 8 sensors-25-00110-f008:**
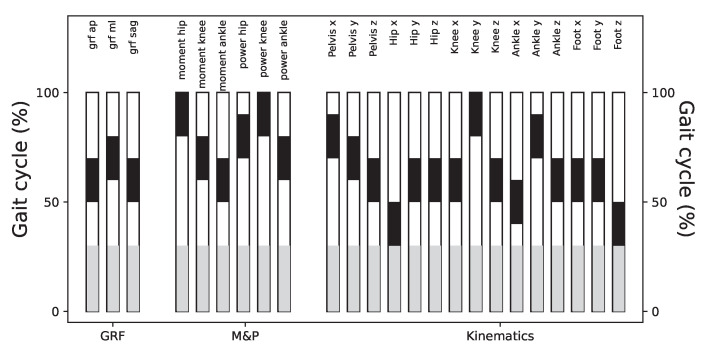
Relevance of different parts of the time series. Each bar reports in black the location of the segment that allows for the obtainment of the best classification score, using a ResNet model. See Figure A10 for full results. The grey part corresponds to the initial 30% of the time series that is discarded in the filtering described in Section 3.1.

**Table 1 sensors-25-00110-t001:** Demographic features and spatiotemporal gait features of healthy children assessed in Dublin and Madrid laboratories.

**Feature**	**Dublin Children** (n = 96)	**Madrid Children** (n = 31)
**Sex** (Female/Male)	48 F (50%)/48 M (50%)	12 F (39%)/19 M (61%)
**Age** (years, median [min, max])	9.00 [4.00, 16.00]	9.00 [5.00, 16.00]
**Weight** (kg, median [min, max])	31.3 [14.6, 94.7]	28.6 [18.5, 75.3]
**Height** (cm, median [min, max])	1.37 [1.05, 1.83]	1.38 [1.00, 1.82]
**Norm. Walking Speed** (s^−1^, median [min, max])	1.74 [1.16, 2.44]	1.49 [1.15, 2.03]
**Cadence** (steps/s, median [min, max])	2.18 [1.70, 2.95]	1.92 [1.56, 2.73]
**Stance Time** (% of gait cycle, median [min, max])	61.2 [57.8, 66.7]	63.3 [60.2, 66.8]

## Data Availability

CRC Dublin Ethics Committee banned the publicationof raw data due to identifiable nature of data. Madrid data are available from previous publication in https://doi.org/10.1371/journal.pone.0192345.s009.

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
