# Peer review of "Deep Learning Unravels Differences Between Kinematic and Kinetic Gait Cycle Time Series from Two Control Samples of Healthy Children Assessed in Two Different Gait Laboratories"

_sensors, 2024, doi:10.3390/s25010110_

Round 1
Reviewer 1 Report
Comments and Suggestions for Authors
I would like to thank the researchers for providing this study. The manuscript is well written and the overall flow reads very well. I agree that a currently open issue is how to performance of machine learning models developed in one laboratory with one dataset translates to another.
I would, however, state already in the Introduction section that the overall aim here is to determine in which gait laboratory data were collected.
Why was this specific research question chosen over any other? For example, many machine learning studies focus on classifying pathological gait from healthy gait, or try to detect gait events from the raw time series data. Would it not make more sense to answer these research questions? I guess the origin of the gait data can easily be resolved using demographics data (age, gender, name of the participant).
Did you mark gait cycles manually? What steps did the post-processing consist of? Did you exclude turns from your analysis?
It is a bit unclear to me what raw time series you provided to the learning algorithms? Did you provide all the joint angle time series at the same time, or were the joint angles from the different joints regarded as different examples?
Did you apply any normalization before providing the time series data to the models?
How did you come up with the models' architectures? Was any hyperparameter tuning performed?
You state that "Some differences are easy to spot by just looking at the time series.". What makes you feel that you then still need a relatively expensive deep learning algorithm to perform the classification?
Author Response
I would, however, state already in the Introduction section that the overall aim here is to determine in which gait laboratory data were collected.
As suggested, we have revisited the introduction to clarify this aspect.
Why was this specific research question chosen over any other? For example, many machine learning studies focus on classifying pathological gait from healthy gait, or try to detect gait events from the raw time series data. Would it not make more sense to answer these research questions? I guess the origin of the gait data can easily be resolved using demographics data (age, gender, name of the participant).
The referee here raises a very interesting question. It is clear that the ultimate goal should be to provide models able to classify control subjects vs. patients, especially towards diagnostic procedures, or more generally to increase our knowledge of a specific pathology. Still, what we here discuss is a prerequisite for that. Suppose we train a DL model achieving the previous goal, using data from one laboratory; if data from a second laboratory are different, also according to a DL architecture, what is the actual generalizability of such model? Can we trust the results it will yield on different data sets, when these data sets are different? Clearly not. Hence, the reason behind this work: understanding how DL models are able to recognize the source laboratory, in spite of shared equipment and assessment approaches; towards creating better guidelines to record homogeneous data sets.
For the sake of clarity, these ideas have been highlighted, both in the introduction and in the conclusions.
Did you mark gait cycles manually?
Yes, we indicated that in the methods.
What steps did the post-processing consist of?
Post-processing was strictly limited to data extraction and data preparation. We included a comment in the methods following the reviewer’s recommendation
Did you exclude turns from your analysis?
Yes, we have now indicated this consideration in the methods.
It is a bit unclear to me what raw time series you provided to the learning algorithms? Did you provide all the joint angle time series at the same time, or were the joint angles from the different joints regarded as different examples?
Each model has been trained with a single set of time series, encoding an individual joint / angular plane; or, in other words, all classifications were univariate in nature. This has been clarified in the manuscript, both in the methods, and as a remainder in the results’ section.
Did you apply any normalization before providing the time series data to the models?
No amplitude normalization has been performed. This was motivated by three considerations:
- We wanted to keep the initial time series as close as possible to what actually recorded; normalizations and other types of pre-processing have been introduced, but as a part of the analyses themselves, see for instance Sec. 3.2.
- Time series already came with acceptable ranges, mostly between (-2, 2); see for instance Figs. 4 and A4.
- Finally, the necessity of time series normalization in Deep Learning classification is a complex topic, and it is not always needed. Actually, at times it is even detrimental: see for instance the results in Sec. 3.3 of Ref. [15] (Crespo et al., 2024).
We have clarified the fact that no normalization has been performed in the text.
How did you come up with the models' architectures? Was any hyperparameter tuning performed?
All DL models considered in this work are standard ones, whose relevance has previously been shown in the literature. For instance, the referee may check Ref. [14], an excellent review in which these models are described in depth and compared. No automatic hyperparameter tuning has been performed; instead, the authors have used configurations that have yielded excellent results in the past, see for instance Ref. [15]. While slightly better classifications may be obtained by tuning some parameters, accuracies are already above 95% most of the time… hence the message would not change.
We have improved the description of these models, to clarify how these and their associated parameters have been obtained.
You state that "Some differences are easy to spot by just looking at the time series.". What makes you feel that you then still need a relatively expensive deep learning algorithm to perform the classification?
Please note that the cited sentence specifically refers to the raw time series, as plotted in Fig. 4; but not to the processed ones, as for instance those in Fig. A4. While a visual inspection can be an initial approach, it presents two drawbacks. Firstly, it is not reliable: the fact that two sets of time series “appear” to be similar does not guarantee that they actually are. Secondly, it cannot provide a numerical quantification of such similarity. If this work would have been based on “time series in Fig. 4 seem different, but now in Fig. A4 they are pretty similar”, we are sure the referees’ comments would not have been very positive!
The importance of the quantitative estimation of the similarity has been clarified in the introduction.
In blue, we have included additional recommendations from the editors.
Reviewer 2 Report
Comments and Suggestions for Authors
This manuscript proposes an enhanced learning model for gait experiment evaluation in different laboratories by exploring preprocessing methods and time series characteristics in Machine Learning. The paper demonstrates a coherent line of reasoning, and the data presented is rich enough to support conclusion. The only question is that Residual Networks is very important in this work, but there is no specific explanation in the article about how this method is established and operates. Please provide a detailed discussion.
Author Response
The only question is that Residual Networks is very important in this work, but there is no specific explanation in the article about how this method is established and operates. Please provide a detailed discussion.
As suggested, we have improved the description of the model and of its implementation in Sec. 2.3. Specifically, we have included a more comprehensive description of the ResNet model, focusing on the organization of the layers; and more details about the specific implementation and training procedure, also describing software aspects.
Round 2
Reviewer 1 Report
Comments and Suggestions for Authors
Dear authors,
many thanks for offering the possibility to review your revision. You have done a tremendous job, and I appreciate that you have taken the time to respond to my previous comments.
The suggested readings are very welcome as well (I was aware of Fawaz et al., 2019, but the Cresop-Otero et al., 2024 is a good read :)).
It would be great if the code and/or models could be shared on a public repository.